# GraphKAN: An Efficient and Interpretable Kolmogorov-Arnold Graph Network for Source Detection

## Abstract

Source detection in graphs offers a viable solution to critical challenges such as rumor tracing. Yet existing GCN-based approaches squander non-embedding parameters and rely on fixed activation functions. We present GraphKAN: An Efficient and Interpretable Kolmogorov–Arnold Graph Network for Source Detection, which capitalizes on Kolmogorov–Arnold Networks (KANs) by assigning learnable activation functions to edge weights. Node features are first diffused through B-spline–based univariate activations, yielding expressive and localized transformations. We further devise a sparsity-aware neighborhood aggregation rooted in community clusters, where edge-level attention is adaptively strengthened through KAN-driven kernel learning. Unlike black-box GCNs, GraphKAN exposes interpretable intermediate representations via its learnable basis functions. Extensive experiments on twelve real-world datasets demonstrate that GraphKAN consistently outperforms state-of-the-art baselines in accuracy, efficiency, and interpretability. Codes will be made public upon paper acceptance.

## 1 Introduction

Source detection on graphs offers a viable solution to pressing societal challenges such as rumor tracing, while simultaneously posing notable mathematical difficulties (Shah & Zaman, 2011; Zhu et al., 2022). Early approaches, including LPSI (Wang et al., 2017), OJC (Zhu et al., 2017), and MLE (Pinto et al., 2012; Yang et al., 2020), rely on source centrality theory (Shah & Zaman, 2011) and maximum likelihood estimation (Cheng et al., 2025) to identify the origin of diffusion. In recent years, with the advancement of deep learning, particularly Graph Convolutional Networks (GCNs) (Kipf & Welling, 2017), researchers embed both node features and social topologies to learn more expressive node representations (Dong et al., 2019; Ling et al., 2022; Wang et al., 2022), achieving new state-of-the-art records.

However, existing GCN-based source detection methods are fundamentally built upon Multi-Layer Perceptrons (MLPs), which leverage the universal approximation theorem to achieve a robust capacity for approximating nonlinear functions (Kiamari et al., 2024). Despite their widespread adoption, MLPs suffer from several notable limitations: 1) the excessive consumption of non-embedding parameters leads to high memory overhead in graph neural networks; 2) the use of fixed activation functions constrains their representational flexibility; and 3) their inherently black-box nature hinders interpretability.

We note that recent progress in the Kolmogorov–Arnold theorem has led to the remarkable development of Kolmogorov–Arnold Networks (KAN) (Liu et al., 2024). Unlike traditional MLPs, where edges carry learnable weights and nodes apply fixed activation functions, KAN assigns learnable activation functions to edges while nodes perform only linear operations on incoming signals. Internally, KAN employs univariate spline functions as activation kernels, which offer strong local transformation capabilities and high accuracy in low-order function approximation. Externally, it captures expressive feature representations through a compositional structure. This design provides KAN with both powerful learning capacity and reduced computational graph complexity compared to MLPs. Additionally, the use of distinct basis functions enhances interpretability. However, KAN cannot be directly applied to source detection tasks. The key challenge lies in incorporating graph

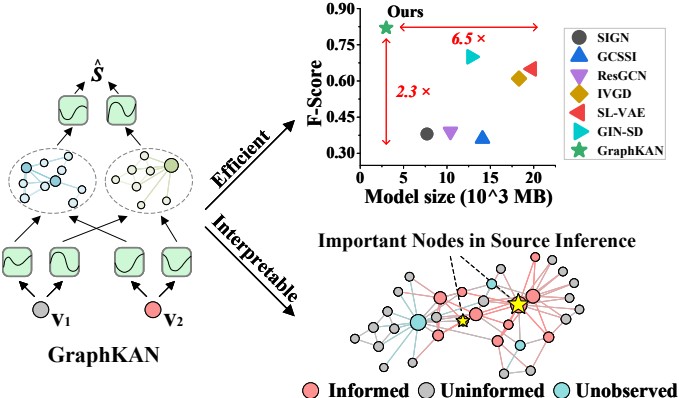

Figure 1: We propose a graph-aware Kolmogorov–Arnold network for source detection (GraphKAN), which achieves significant improvements in accuracy and efficiency over baseline methods, while also enhancing interpretability.

topology to design efficient information aggregation mechanisms, which is essential for adapting KAN to this domain.

In this paper, we propose the GraphKAN: An Efficient and Interpretable Kolmogorov-Arnold Graph Network for Source Detection, which harnesses the expressive power of learnable activation functions within a graph-based inference framework. Specifically, node features are first propagated via univariate B-spline-based activation functions, enabling localized and expressive nonlinear transformations. To effectively integrate structural information while preserving computational scalability, we design a sparsity-aware neighborhood aggregation strategy grounded in community-based clustering. This mechanism adaptively enhances edge-level attention weights through KAN's learnable kernel functions, allowing for flexible and topology-sensitive message passing. Furthermore, we provide interpretable insights into the relative importance of different nodes in the source detection process through the analysis of learned activation patterns. We evaluate our approach on 12 real-world datasets. Extensive experimental results demonstrate that GraphKAN surpasses state-of-the-art techniques in terms of accuracy, efficiency, and interpretability, establishing new baselines.

Overall, our contributions are summarized as:

- We propose GraphKAN, a novel graph learning framework for source detection that leverages the Kolmogorov–Arnold representation to enable expressive, localized, and interpretable nonlinear modeling within graph neural networks.

- We design a sparsity-aware neighborhood aggregation mechanism that integrates graph topology via community-based clustering and adaptively enhances edge attention through KAN's learnable kernels, enabling efficient and structure-aware message passing.

- We show by extensive experiments on 12 real-world datasets that GraphKAN consistently outperforms state-of-the-art methods in accuracy, efficiency, and interpretability, establishing new performance baselines.

## 2 RELATED WORK

### 2.1 SNAPSHOT-BASED MULTI-SOURCE DETECTION

In recent years, snapshot-based approaches have gained popularity for multi-source detection due to their ease of access and ability to capture essential information such as user states and network topology (Cheng et al., 2024b). Based on source centrality theory (Prakash et al., 2012; Shah & Zaman, 2011), LPSI identifies locally prominent nodes via label propagation (Wang et al., 2017), EPA estimates infection times iteratively (Ali et al., 2019), and OJC optimizes Jordan centrality (Zhu et al., 2017). While computationally efficient, these methods struggle to handle the complexity of user attributes in real-world networks (Cheng et al., 2024a). Graph neural network-based approaches have emerged as powerful alternatives (Bao et al., 2024). GCNSI (Dong et al., 2019) and SIGN (Li

et al., 2021) incorporate user states and related attributes as input features for node classification, while GCSSI targets wavefront nodes (Dong et al., 2022). From a structural modeling perspective, ResGCN enhances information propagation through residual connections (Shah et al., 2020). However, these methods fall short in capturing the underlying dynamics of information diffusion. To address this, IVGD (Wang et al., 2022) and SL-VAE (Ling et al., 2022) incorporate graph diffusion processes to learn diverse propagation patterns. Despite these advances, existing GCN-based methods fundamentally rely on MLP backbones, which suffer from limited efficiency, flexibility, and interpretability due to their overparameterized structures, fixed activations, and black-box nature.

## 2.2 KOLMOGOROV–ARNOLD NETWORKS (KAN)

The Kolmogorov–Arnold representation theorem establishes that any multivariate continuous function within a bounded domain can be represented as a finite superposition of univariate functions in a binary composition (Kolmogorov, 1957). Although earlier studies have attempted to leverage this theoretical foundation for machine learning (Sprecher & Draghici, 2002; Fakhoury et al., 2022; Montanelli & Yang, 2020), they are constrained to networks of fixed depth (2) and width ($2n+1$). By generalizing the theorem, Kolmogorov–Arnold Networks (KAN) extend the representation to arbitrary depth and width, enabling seamless integration into contemporary deep learning pipelines (Liu et al., 2024). KAN employs univariate spline-based activation functions with strong local transformation capacity, combined with a compositional architecture, achieving both high approximation accuracy and interpretability. These properties make KAN a promising alternative to traditional MLPs. However, KAN cannot be directly applied to source detection tasks. The primary challenge lies in incorporating graph structural information to design effective aggregation mechanisms.

## 3 PROBLEM FORMULATION

**Preliminary on Social Networks.** The social network in physical world can be abstracted as graph $G = (V, E)$, where $V = \{v_1, v_2, \cdots, v_n\}$ denotes the set of nodes representing users, and $E = \{(v_i, v_j) \mid v_i, v_j \in V, i \neq j\}$ denotes the set of edges representing social interactions. Each node $v_i \in V$ may be associated with a feature vector $\boldsymbol{X}_i \in \mathbb{R}^d$, capturing user attributes such as profile information and activity status. The overall topology captures the structural properties of the underlying social system. We denote by $\mathcal{N}(i)$ the set of neighbors of node $v_i$ and $\boldsymbol{A}$ ($\boldsymbol{A}_{ij} \in \{0, 1\}^{n \times n}$) the adjacency matrix.

**Propagation Process on Social Networks.** Information diffusion on social networks evolves over time $t$. At $t = 0$, a subset of sources $s$ transitions from uninformed to informed, initiating the cascade; for $t > 0$, each informed user independently forwards to neighbors with a personal forwarding probability $p$. Canonical diffusion models (SI, SIR, IC, and LT) simulate this process (Battiston et al., 2020; de Arruda et al., 2020). Accordingly, the propagation is represented by time-indexed snapshots $\{G'_t\}_{t \geq 0}$, where each $G'_t$ partitions nodes into informed $G_+$ and uninformed $G_-$.

**Source Detection in Graphs.** Once the informed fraction attains a prespecified threshold $\delta \in (0, 1)$, we obtain a snapshot $G'$ comprising the topology $T$, user infection states $U$, and propagation information $P$. Formally, the problem is defined as:

$$\hat{s} = f(G'(T, U, P)), \tag{1}$$

where $f(\cdot)$ represents the source detection algorithm, and $\hat{s}$ denotes the set of detected sources.

**GCN-based vs. KAN-based Approaches.** The core difference between GCN-based and KAN-based source detection methods lies in their nonlinear transformation. GCN-based methods model the detection function $f(\cdot)$ using MLP backbones with fixed activations $\sigma(\cdot)$ and non-embedding weights $\boldsymbol{W}$, where information is aggregated via:

$$\boldsymbol{X}_i^{(l+1)} = \sigma \left( \sum_{j \in \mathcal{N}(i)} \boldsymbol{W}^{(l)} \boldsymbol{X}_j^{(l)} \right). \tag{2}$$

While KAN-based approach replaces fixed activations with learnable B-spline functions $\phi(\cdot)$:

$$\boldsymbol{X}_i^{(l+1)} = \mathcal{A}_{\text{GraphKAN}} \left( \left\{ \phi_{ij} \left( \boldsymbol{X}_j^{(l)} \right) \mid j \in \mathcal{N}(i) \right\} \right), \tag{3}$$

where $\mathcal{A}_{\text{GraphKAN}}$ denotes our proposed aggregator.

## 4 METHOD

In this section, we present GraphKAN, a dedicated framework designed to address the challenges Above. Specifically, we design the **KAN-based Node Representation** module to address Challenge 1), and propose a **Community-guided Sparse Aggregation** mechanism for Challenge 2).

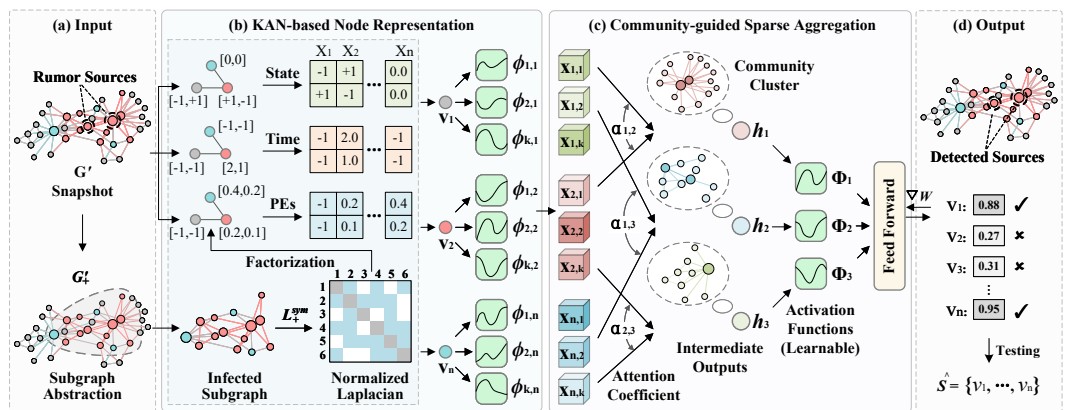

Figure 2: GraphKAN framework. (a) Input: snapshot with informed, uninformed, missing-state nodes. (b) KAN representation: embed state, timestamp, positional encoding via learnable B-spline activations. (c) Community-guided sparse aggregation: aggregate kernel-enhanced features with sparsity-aware, kernel-adaptive attention. (d) Output: softmax yields source probabilities.

### 4.1 KAN-BASED NODE REPRESENTATION

To facilitate effective comparison and representation learning, several key attributes captured in the snapshot $G'_t$ are embedded as node features.

**State Information $X_i^1$.** The user state reflects whether a user has participated in the propagation. In the snapshot $G'$, users fall into several subsets: $G_+$ (informed), $G_-$ (uninformed), and $\Pi$, which represents users with missing state information due to privacy constraints or incomplete observations. The corresponding state feature $X_i^1$ is defined as:

$$X_i^1 = \begin{cases} +1, & v_i \in G_+ \\ -1, & v_i \in G_- \\ 0, & v_i \in \Pi. \end{cases} \tag{4}$$

**Propagation Information $X_i^2$.** To model diffusion dynamics, each node's participation timestamp is incorporated as the temporal feature. For informed nodes $v_i \in G_+$, we record the time $t_i$ at which the node was first influenced by the rumor. For uninformed nodes $v_i \in G_-$ or nodes with missing information $v_i \in \Pi$, a default value (-1) is assigned. The resulting temporal feature $X_i^2$ is defined as:

$$X_i^2 = \begin{cases} t_i, & v_i \in G_+ \\ -1, & \text{otherwise}. \end{cases} \tag{5}$$

**Positional Encodings $X_i^3$.** In scenarios with partially missing node information, relative structural positions provide key signals for modeling diffusion and enhancing global propagation. However, existing GCN-based models learn representations with invariant node positions (Srinivasan & Ribeiro, 2019). To address this, we adopt Laplacian positional encodings (Dwivedi et al., 2020) as structural features due to their strong generalization capability.

Motivated by source centrality theory, rather than computing positional encodings over the entire graph, we first extract an infection subgraph and then perform position encoding within this local structure, thereby emphasizing the topological centrality of potential sources. For uninformed nodes $v_i$, the subgraph extraction process is formulated as:

$$A_+ = J_{i,n} \cdot A \cdot J_{i,n}^T, \tag{6}$$

where $\boldsymbol{A} \in \mathbb{R}^{n \times n}$ denotes the adjacency matrix of the full graph $G'$, and $\boldsymbol{A}_+ \in \mathbb{R}^{(n-1) \times (n-1)}$ is the adjacency matrix after removing node $v_i$. $\boldsymbol{J}_{i,n} \in \mathbb{R}^{(n-1) \times n}$ is a selection matrix derived from the identity matrix by deleting the $i$-th row, which serves to remove node $v_i$ and its associated edges.

After extracting the infection-induced subgraph, the symmetrically normalized Laplacian matrix is computed as:

$$\boldsymbol{L}_+^{sym} = \boldsymbol{I} - \boldsymbol{D}_+^{-1/2} \boldsymbol{A}_+ \boldsymbol{D}_+^{-1/2}, \tag{7}$$

where $\boldsymbol{D}_+$ is the degree matrix associated with $\boldsymbol{A}_+$. The normalized Laplacian $\boldsymbol{L}_+^{sym}$ can be further decomposed via eigendecomposition as:

$$\triangle_{\boldsymbol{L}_+^{sym}} = \Gamma^T \lambda \Gamma, \tag{8}$$

where $\lambda$ is a diagonal matrix of eigenvalues and $\Gamma$ contains the corresponding eigenvectors. We select the eigenvectors associated with the $r$-smallest nontrivial eigenvalues to form the positional encoding matrix ($r \ll n$), yielding the final positional feature $\boldsymbol{X}_i^3 \in \mathbb{R}^r$:

$$\boldsymbol{X}_i^3 = \begin{cases} \Gamma_i, & v_i \in G_+ \cup \Pi \\ -1, & \text{otherwise.} \end{cases} \tag{9}$$

To ensure compatibility with KAN's localized activation structure and facilitate the capture of node-level propagation patterns with improved interpretability, we apply a transformation to the concatenated features such that each node representation serves as a basic computational unit for KAN:

$$\boldsymbol{X}_i = \boldsymbol{W} \cdot \left[ \|_{x=1}^3 \boldsymbol{X}_i^x \right] + \mathbf{b}. \tag{10}$$

**Preliminary on KAN.** A KAN layer is characterised by a matrix of univariate functions $\Phi^{(l)} = \{\phi_{j,i}^{(l)}\}, i = 1, \ldots, n_l, j = 1, \ldots, n_{l+1}$, where $n_l$ and $n_{l+1}$ denote the input and output widths, respectively. For a more intuitive illustration, the layer-wise transformation can be represented as:

$$x_j^{(l+1)} = \sum_{i=1}^{n_l} \phi_{j,i}^{(l)} \big( x_i^{(l)} \big), \qquad j = 1, \ldots, n_{l+1}, \tag{11}$$

with each $\phi_{j,i}^{(l)}$ instantiated as a learnable B-spline. Stacking such layers preserves universal approximation while endowing KAN with depth and gradient-based trainability.

**Kernel-driven Node Feature Diffusion.** Given the initial node embedding $\boldsymbol{X}_i \in \mathbb{R}^d$ and KAN foundations, we compute the latent projection and spline responses:

$$\xi_i = \mathbf{a}^\top \boldsymbol{X}_i, \tag{12}$$
$$g_{i,r} = B_r(\xi_i; \mu_r, \sigma_r), \; r = 1, \ldots, k, \tag{13}$$

where $\mathbf{a} \in \mathbb{R}^d$ is a trainable vector, and the set $\{B_r\}$ comprises $k$ learnable cubic B-spline kernels with centres $\mu_r$ and widths $\sigma_r$. The response vector $\mathbf{g}_i = [g_{i,1}, \ldots, g_{i,k}]^\top$ constitutes a kernel-based non-linear enhancement of node features, capturing expressive representations that are preserved for subsequent attention and aggregation.

## 4.2 COMMUNITY-GUIDED SPARSE AGGREGATION

In this subsection, we detail the components of GraphKAN that enable topology-sensitive message passing through sparse aggregation and adaptive attention.

**Community-aware Sparse Message Passing with Adaptive Attention.** Information exchange in social graphs is typically more intensive within communities. We therefore derive a community mask $\mathbf{P}_{\text{comm}} \in \{0,1\}^{n \times n}$ using the Louvain algorithm (Traag et al., 2019) and restrict message passing to its non-zero pattern. To further reduce complexity, we retain only the top-$k_{\max}$ high-degree nodes within each community as designated message-passing targets for all other members:

$$(\mathbf{P}_{\text{comm}})_{ij} = \begin{cases} 1, & \text{if } v_j \in \text{top-}k_{\max}(\mathcal{C}(v_i)) \\ 0, & \text{otherwise,} \end{cases} \tag{14}$$

where $\mathcal{C}(v_i)$ denotes the community containing node $v_i$.

Beyond conventional graph attention based on feature similarity (Ma et al., 2024), we introduce kernel-driven adaptive attention to endow edges with learnable, data-driven weights. To promote independent and diverse feature extraction, multiple attention channels are employed. For each channel $q = 1, \ldots, m$, the unnormalized edge score and normalized attention are computed as:

$$e_{ij}^{(q)} = \mathbf{a}_q^\top [\mathbf{z}_i \| \mathbf{z}_j] + \sum_{r=1}^{k} \beta_{q,r} (g_{i,r} + g_{j,r}), \tag{15}$$

$$\alpha_{ij}^{(q)} = \frac{\exp\big(\sigma(e_{ij}^{(q)})\big)}{\sum_{j' \in \mathcal{N}_i^{\mathrm{comm}}} \exp\big(\sigma(e_{ij'}^{(q)})\big)}, \tag{16}$$

where $\mathbf{a}_q \in \mathbb{R}^{2d}$ is the standard GAT vector, $\beta_{q,r}$ is a learnable kernel weight, $\sigma(\cdot)$ denotes LeakyReLU, and $\mathcal{N}_i^{\mathrm{comm}}$ is the neighbor set defined by $\mathbf{P}_{\mathrm{comm}}$.

**Channel-wise Aggregation and Non-linearity.** Following the calculation of attention coefficients, kernel responses are diffused and aggregated per channel:

$$\tilde{g}_{i,r}^{(q)} = \sum_{j \in \mathcal{N}_i^{\mathrm{comm}}} \alpha_{ij}^{(q)} g_{j,r}, \quad s_i^{(q)} = \sum_{r=1}^{k} w_{q,r} \tilde{g}_{i,r}^{(q)}, \tag{17}$$

$$\mathbf{S}_i = [s_i^{(1)}, \ldots, s_i^{(m)}]^\top, \tag{18}$$

$w_{q,r}$ denotes a learnable channel weight. To promote diversity, $\mathbf{S}_i$ is processed by a point-wise non-linearity:

$$\mathbf{U}_i = \Phi(\mathbf{S}_i) = B_m(s_i^{(m)}), \qquad \mathbf{U}_i \in \mathbb{R}^m, \tag{19}$$

which forms the input to next GraphKAN layers. The final binary-classification logits are passed through a softmax function to produce the estimated source probabilities:

$$\hat{\mathbf{p}}_i = \frac{\exp\big(\boldsymbol{W}_{\mathrm{cls}} \mathbf{U}_i + \mathbf{b}_{\mathrm{cls}}\big)}{\sum_{c'=0}^{1} \exp\big((\boldsymbol{W}_{\mathrm{cls}} \mathbf{U}_i + \mathbf{b}_{\mathrm{cls}})_{c'}\big)}, \tag{20}$$

$$\hat{\mathbf{p}}_i = \big[\hat{p}_{i,1}, \hat{p}_{i,0}\big]^\top, \tag{21}$$

$\hat{p}_{i,1}$ and $\hat{p}_{i,0}$ are the estimated probabilities that $v_i$ is a rumor source and a non-source, respectively.

**Interpretability.** Since each node is processed via univariate kernels and edge-adaptive attention, variables such as kernel responses $g_{i,r}$, attention weights $\beta_{q,r}$ and community mask $\mathbf{P}_{\mathrm{comm}}$ can be inspected. This transparency enables fine-grained analysis of each node's influence on source inference and paves the way for interpretable graph learning.

## 4.3 OPTIMIZATION AND TRAINING

In rumor-spreading snapshots the number of sources is typically negligible compared with that of non-sources, which can bias the model toward negative predictions. To compensate for this class imbalance we introduce a weighting factor:

$$\omega = \frac{n - |s|}{|s|}, \tag{22}$$

where $n$ is the total number of nodes in the snapshot and $|s|$ is the number of labelled sources. All source samples are multiplied by $\omega$, whereas non-source samples keep unit weight, yielding equal aggregate weight for the two classes.

**Objective function.** Given $\hat{\mathbf{p}}_i = \big[\hat{p}_{i,1}, \hat{p}_{i,0}\big]^\top$ as the softmax output for node $v_i$ and its ground-truth label $y_i \in \{1, 0\}$, the weighted cross-entropy loss over a snapshot is:

$$\mathcal{L}_{\mathrm{CE}} = -\frac{1}{n} \sum_{i=1}^{n} \big(\omega\, y_i \log \hat{p}_{i,1} + (1 - y_i) \log \hat{p}_{i,0}\big). \tag{23}$$

To prevent over-fitting we add an $\ell_2$ penalty on all trainable parameters $\Theta$:

$$\mathcal{L} = \mathcal{L}_{\mathrm{CE}} + \varepsilon \|\Theta\|_2^2, \tag{24}$$

where $\varepsilon > 0$ is the regularization coefficient. The model is optimized end-to-end until convergency.

## 5 EXPERIMENTS

### 5.1 EXPERIMENTAL SETTINGS

**Datasets.** We conduct experiments on twelve real-world datasets, including six static networks: Football (Girvan & Newman, 2002), Jazz (Gleiser & Danon, 2003), Facebook (Leskovec & Mcauley, 2012), LastFM (Rozemberczki & Sarkar, 2020), Enron (Klimt & Yang, 2004), and Github (Rozemberczki et al., 2021); and six cascade datasets: Christianity (Sankar et al., 2020), Meme-tracker (Leskovec et al., 2009), Android (Sankar et al., 2020), Twitter (Hodas & Lerman, 2014), Douban (Zhong et al., 2012), and Weibo (Cao et al., 2017). The static networks differ in size, degree, and clustering, while the cascade datasets capture time-resolved user interactions and propagation. Together, they support comprehensive evaluation across diverse network settings.

**Baselines.** Different types of methods are selected as baselines: centrality-based methods such as LPSI (Wang et al., 2017) and EPA (Ali et al., 2019), GCN-based methods leveraging user states including GCNSI (Dong et al., 2019), SIGN (Li et al., 2021), GCSSI (Dong et al., 2022) and Res-GCN (Shah et al., 2020), and propagation-aware models such as IVGD (Wang et al., 2022), SL-VAE (Ling et al., 2022) and GIN-SD (Cheng et al., 2024b).

**Implementation.** For Networks 1–6, we simulate diffusion using the IC model: 3% of nodes are randomly designated as sources, each informed node forwards with probability $p_i \sim U(0, 0.5)$, and snapshots are captured when 30% of nodes become informed. To emulate missing data, 2% of node states are masked. For Datasets 7–12, the first user in each cascade is treated as the ground-truth source. Samples are split 8:2 into training and test sets. GraphKAN is instantiated with two KAN-Mix layers, each with $m = 3$ attention channels and $k = 4$ B-spline kernels. Community-aware sparsity is enforced by selecting the top-5 high-degree nodes per community for message passing. The model is trained via Adam with a learning rate of $10^{-3}$ and weight decay of $10^{-5}$. All experiments run on a workstation equipped with four NVIDIA RTX 3090Ti GPUs.

**Metrics.** We evaluate performance using three standard metrics: accuracy (ACC), F1, and area under the ROC curve (AUC). ACC measures the proportion of nodes that are correctly classified as source or non-source. F1 balances precision and recall, where precision is $|\hat{s} \cap s|/|\hat{s}|$ and recall equals $|\hat{s} \cap s|/|s|$ with $\hat{s}$ denoting the predicted source set and $s$ the ground-truth set. AUC quantifies the model's classification capability across all decision thresholds. These three complementary metrics jointly provide a comprehensive and nuanced view of overall model performance.

### 5.2 PERFORMANCE ANALYSIS

**Comparison with State-of-the-art Methods.** Table 1 reports the performance of GraphKAN and all baselines. Several observations arise. First, ACC are consistently higher than F1 for every method. This gap reflects the pronounced class imbalance between source and non-source nodes: centrality-based models (LPSI, EPA) and state-driven GCN variants (GCNSI, SIGN, GCSSI) are particularly susceptible to this skew, resulting in low precision and thus depressed F1. Second, learning-based approaches that integrate multiple node features generally surpass purely centrality-oriented heuristics. Within this group, propagation-aware models (IVGD, SL-VAE) further improve performance by explicitly capturing temporal diffusion patterns. GIN-SD attains the strongest baseline results by additionally handling missing-state nodes. Finally, GraphKAN outperforms all competitors on every dataset, it delivers 15%–25% absolute gains, and achieves up to a two-fold improvement over centrality-based methods. These gains stem from two designs: (i) spline-based node representations that capture fine-grained nonlinear cues, and (ii) community-aware sparse aggregation with kernel-adaptive attention that models heterogeneous propagation paths.

**Visualization.** To offer an intuitive comparison, we visualize the predicted sources of GraphKAN alongside those of representative baseline methods on the Jazz network. As shown in Fig. 3, GraphKAN correctly identifies a greater number of true sources compared to competing approaches.

**Computational Efficiency.** Model sizes (memory footprints) are reported in Table 2, their trade-off with F1-score is visualised in Fig. 4, and runtimes are summarised in Table 3. Centrality-based approaches exhibit the smallest footprints but also the lowest F1. GCN-based models incur substantially larger sizes due to heavy non-embedding parameters and dense attention. In contrast, GraphKAN shifts expressiveness to learnable activation kernels and employs sparsity-aware ag-

| Datasets | Metrics | Methods | | | | | | | | | |
|---|---|---|---|---|---|---|---|---|---|---|---|
| | | LPSI | EPA | GCNSI | SIGN | GCSSI | ResGCN | IVGD | SL-VAE | GIN-SD | Ours |
| Football | ACC | 0.81 | 0.82 | 0.81 | 0.82 | 0.78 | 0.82 | 0.85 | 0.84 | 0.88 | **0.96** |
| | F1 | 0.31 | 0.33 | 0.25 | 0.43 | 0.41 | 0.45 | 0.68 | 0.66 | 0.71 | **0.80** |
| | AUC | 0.85 | 0.83 | 0.83 | 0.83 | 0.82 | 0.84 | 0.86 | 0.83 | 0.86 | **0.97** |
| Jazz | ACC | 0.83 | 0.80 | 0.81 | 0.84 | 0.79 | 0.82 | 0.83 | 0.83 | 0.85 | **0.95** |
| | F1 | 0.30 | 0.31 | 0.23 | 0.40 | 0.37 | 0.39 | 0.61 | 0.62 | 0.68 | **0.75** |
| | AUC | 0.84 | 0.84 | 0.82 | 0.81 | 0.81 | 0.81 | 0.85 | 0.82 | 0.84 | **0.94** |
| Facebook | ACC | 0.85 | 0.81 | 0.77 | 0.82 | 0.83 | 0.84 | 0.83 | 0.82 | 0.85 | **0.95** |
| | F1 | 0.24 | 0.25 | 0.11 | 0.44 | 0.41 | 0.43 | 0.67 | 0.65 | 0.69 | **0.81** |
| | AUC | 0.81 | 0.80 | 0.75 | 0.81 | 0.84 | 0.85 | 0.85 | 0.83 | 0.86 | **0.94** |
| LastFM | ACC | 0.86 | 0.81 | 0.78 | 0.83 | 0.81 | 0.82 | 0.84 | 0.81 | 0.89 | **0.93** |
| | F1 | 0.22 | 0.20 | 0.09 | 0.41 | 0.39 | 0.40 | 0.62 | 0.61 | 0.69 | **0.75** |
| | AUC | 0.82 | 0.79 | 0.75 | 0.85 | 0.79 | 0.83 | 0.85 | 0.84 | 0.90 | **0.92** |
| Enron | ACC | 0.84 | 0.83 | 0.74 | 0.79 | 0.80 | 0.81 | 0.83 | 0.82 | 0.85 | **0.94** |
| | F1 | 0.20 | 0.23 | 0.07 | 0.39 | 0.37 | 0.39 | 0.59 | 0.58 | 0.67 | **0.74** |
| | AUC | 0.83 | 0.81 | 0.76 | 0.82 | 0.78 | 0.83 | 0.85 | 0.84 | 0.88 | **0.92** |
| Github | ACC | 0.81 | 0.79 | 0.73 | 0.82 | 0.83 | 0.82 | 0.84 | 0.82 | 0.87 | **0.92** |
| | F1 | 0.19 | 0.17 | 0.08 | 0.35 | 0.36 | 0.38 | 0.60 | 0.61 | 0.61 | **0.72** |
| | AUC | 0.82 | 0.81 | 0.71 | 0.81 | 0.84 | 0.83 | 0.86 | 0.84 | 0.85 | **0.94** |
| Christianity | ACC | 0.82 | 0.78 | 0.81 | 0.80 | 0.81 | 0.82 | 0.81 | 0.84 | 0.87 | **0.94** |
| | F1 | 0.23 | 0.24 | 0.12 | 0.37 | 0.38 | 0.41 | 0.56 | 0.54 | 0.68 | **0.75** |
| | AUC | 0.83 | 0.82 | 0.75 | 0.81 | 0.79 | 0.83 | 0.83 | 0.81 | 0.91 | **0.93** |
| Memetracker | ACC | 0.85 | 0.84 | 0.77 | 0.79 | 0.81 | 0.82 | 0.84 | 0.84 | 0.87 | **0.92** |
| | F1 | 0.22 | 0.24 | 0.13 | 0.35 | 0.34 | 0.38 | 0.54 | 0.55 | 0.64 | **0.71** |
| | AUC | 0.87 | 0.82 | 0.76 | 0.81 | 0.80 | 0.83 | 0.83 | 0.82 | 0.88 | **0.91** |
| Android | ACC | 0.86 | 0.81 | 0.74 | 0.81 | 0.82 | 0.82 | 0.83 | 0.84 | 0.89 | **0.95** |
| | F1 | 0.21 | 0.20 | 0.10 | 0.38 | 0.36 | 0.39 | 0.52 | 0.54 | 0.73 | **0.82** |
| | AUC | 0.84 | 0.80 | 0.80 | 0.78 | 0.83 | 0.84 | 0.85 | 0.85 | 0.92 | **0.96** |
| Twitter | ACC | 0.85 | 0.82 | 0.73 | 0.78 | 0.81 | 0.82 | 0.81 | 0.85 | 0.84 | **0.90** |
| | F1 | 0.20 | 0.23 | 0.09 | 0.31 | 0.33 | 0.36 | 0.49 | 0.47 | 0.52 | **0.63** |
| | AUC | 0.88 | 0.84 | 0.78 | 0.75 | 0.79 | 0.84 | 0.80 | 0.84 | 0.86 | **0.91** |
| Douban | ACC | 0.86 | 0.81 | 0.79 | 0.81 | 0.82 | 0.84 | 0.81 | 0.82 | 0.88 | **0.89** |
| | F1 | 0.18 | 0.21 | 0.15 | 0.32 | 0.31 | 0.35 | 0.48 | 0.50 | 0.54 | **0.65** |
| | AUC | 0.84 | 0.80 | 0.82 | 0.79 | 0.81 | 0.83 | 0.83 | 0.83 | 0.86 | **0.90** |
| Weibo | ACC | 0.84 | 0.81 | 0.81 | 0.77 | 0.80 | 0.82 | 0.81 | 0.82 | 0.86 | **0.88** |
| | F1 | 0.17 | 0.19 | 0.13 | 0.30 | 0.33 | 0.36 | 0.50 | 0.52 | 0.53 | **0.67** |
| | AUC | 0.85 | 0.82 | 0.80 | 0.80 | 0.78 | 0.84 | 0.84 | 0.83 | 0.87 | **0.91** |

Networks 1-6 (rows Football–Github); Datasets 7-12 (rows Christianity–Weibo)

Table 1: Performance comparison of all evaluated methods across the twelve datasets, with the best results being highlighted.

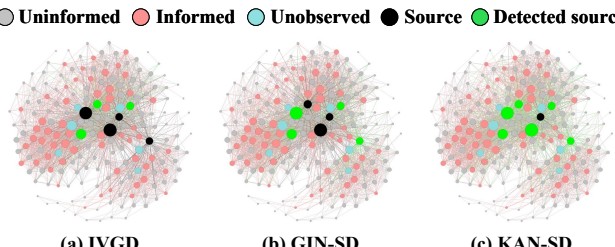

● Uninformed ● Informed ● Unobserved ● Source ● Detected source

(a) IVGD     (b) GIN-SD     (c) KAN-SD

Figure 3: Visualization of source detection results on Jazz.

gregation, achieving the highest detection accuracy with a markedly smaller parameter budget and comparable runtime, demonstrating its practical efficiency.

| Datasets | LPSI | EPA | GCNSI | SIGN | GCSSI | ResGCN | IVGD | SL-VAE | GIN-SD | Ours |
|---|---|---|---|---|---|---|---|---|---|---|
| Facebook | 0.715 | 2.405 | 6.723 | 8.156 | 12.148 | 10.326 | 19.240 | 18.248 | 15.042 | 1.065 |
| Enron | 1.921 | 5.148 | 12.533 | 13.648 | 19.427 | 21.529 | 28.629 | 26.318 | 24.215 | 3.593 |
| Android | 0.748 | 3.549 | 4.647 | 7.720 | 14.098 | 10.378 | 18.262 | 19.764 | 12.834 | 1.543 |
| Douban | 1.644 | 4.078 | 10.194 | 12.013 | 17.468 | 19.267 | 26.052 | 23.594 | 21.480 | 6.963 |

Table 2: Model size ($10^3$ MB) comparison across all baseline methods on benchmark datasets.

**Interpretability.** To quantify each node's contribution in source inference, we extract a composite importance score from the first GraphKAN layer by aggregating its attention-weighted B-spline responses. Nodes ranked by this score show that over 80% of true sources occupy the top positions on the Jazz, Facebook, and Christianity networks (Fig. 5 and Table 4). This strong alignment provides

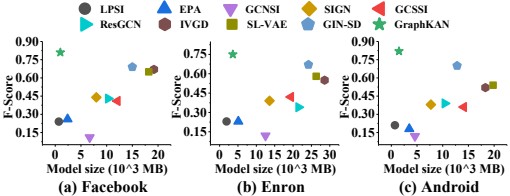

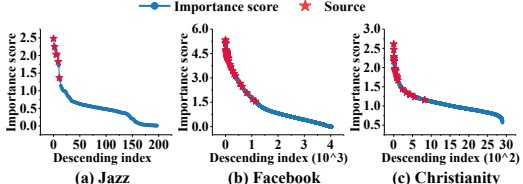

Figure 4: Model size vs. F-score for all methods. GraphKAN attains the best performance.

Figure 5: Importance scores in descending order, with red stars marking the true sources.

| Datasets | LPSI | GCNSI | IVGD | GIN-SD | Ours |
|---|---|---|---|---|---|
| Facebook | 1.346 | 1.545 | 1.816 | 1.648 | 1.503 |
| Enron | 3.914 | 3.821 | 4.259 | 4.148 | 3.745 |
| Android | 2.254 | 2.106 | 2.619 | 2.352 | 2.041 |

Table 3: Runtime ($10^3$ s) of selected methods.

| Method | Jazz | Facebook | Christianity |
|---|---|---|---|
| GraphKAN | 0.925 | 0.854 | 0.836 |

Table 4: Fraction of true sources ranked as top-scoring across datasets.

a clear, quantitative measure of node influence, enabling transparent and fine-grained interpretation of the model's decision process.

| Methods | Facebook | Enron | Android | Twitter |
|---|---|---|---|---|
| w/o P | 0.459 | 0.412 | 0.384 | 0.367 |
| w/o PEs | 0.751 | 0.706 | 0.748 | 0.597 |
| w/ GCN | 0.567 | 0.534 | 0.528 | 0.517 |
| w/ GAT | 0.712 | 0.685 | 0.703 | 0.515 |
| w/ $A_1$ | 0.594 | 0.583 | 0.615 | 0.475 |
| w/ Single_C | 0.728 | 0.634 | 0.724 | 0.549 |
| w/o KAN_Att | 0.674 | 0.548 | 0.613 | 0.508 |
| GraphKAN | **0.812** | **0.740** | **0.819** | **0.634** |

Table 5: Performance of different GraphKAN variants.

## 5.3 ABLATION STUDY AND OTHER ANALYSES

**Effects of Node Representation.** We first remove propagation features (row w/o P), which causes a substantial F1 decline across all datasets, confirming that temporal signals are critical for modelling source dynamics. Removing positional encodings (w/o PEs) also degrades performance, indicating that relative structural position helps mitigate missing-state nodes and improves discrimination.

**Effects of Existing Graph Learning Models.** Replacing GraphKAN's kernel-driven aggregation with GCN sharply reduces performance, revealing GCN's difficulty in modeling complex diffusion. GAT improves on GCN by weighting node importance, yet both use fixed activations, limiting expressiveness and final source-detection accuracy.

**Effects of Sparse Aggregation and Adaptive Attention.** Using raw adjacency $A_1$ impairs performance, confirming community-sparsity aids rumor modeling. Collapsing channels to single (w/ Single_C) worsens results, showing multi-channel prevents collapse. Removing kernel-enhanced attention (w/o KAN_Att) also reduces performance, proving feature-only attention is inadequate. Combining all components, GraphKAN achieves the best accuracy.

## 6 CONCLUSION

This study proposes an accurate, efficient, and interpretable framework for rumor source detection. The key idea lies in constructing robust node representations by integrating heterogeneous features and positional encodings to alleviate incomplete observations, performing kernel-driven feature diffusion with learnable B-spline activations coupled with community-aware sparse aggregation and kernel-adaptive attention, and enabling transparent analysis of internal graph learning behaviour through inspection of activation kernels and attention weights. Extensive experiments on twelve datasets demonstrate that GraphKAN consistently outperforms strong baselines across all metrics. We hope this study inspires further research on effective and interpretable graph learning for diffusion-driven inference tasks.

ETHICS STATEMENT

This study does not involve human subjects, sensitive data, or any practice that may raise ethical concerns outlined in the ICLR Code of Ethics. The datasets used are publicly available, and our method poses no foreseeable societal or security risks. No conflicts of interest, sponsorship influence, or fairness issues were encountered during this research.

REPRODUCIBILITY STATEMENT

For reproducibility, we provide detailed hyperparameter settings, training protocols, and dataset descriptions in the Experimental Settings section. Upon acceptance, we will release the complete implementation, including data preprocessing scripts, model training, and evaluation routines.

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

## A  APPENDIX

### A.1  STATEMENT ON LLM USAGE

LLMs were used only for language polishing (grammar, phrasing, clarity), not for ideation, study design, data analysis, or substantive content generation.

### A.2  KOLMOGOROV–ARNOLD REPRESENTATION THEOREM.

The Kolmogorov–Arnold theorem provides a foundational result in approximation theory, showing that any continuous function on a high-dimensional box can be reduced to a combination of univariate functions. Formally, for:

$$f : [0,1]^n \to \mathbb{R}, \tag{25}$$

there exist continuous outer functions $\{\Phi_q\}_{q=1}^{2n+1}$ and inner functions $\{\phi_{q,p}\}_{1 \leq p \leq n,\, 1 \leq q \leq 2n+1}$:

$$f(x_1, \ldots, x_n) = \sum_{q=1}^{2n+1} \Phi_q\Big( \sum_{p=1}^{n} \phi_{q,p}(x_p) \Big). \tag{26}$$

This decomposition has several remarkable implications:

- The number of outer terms, $2n + 1$, depends only on the input dimension and not on the complexity of $f$.

- Each inner sum $\sum_p \phi_{q,p}(x_p)$ effectively projects the vector $\mathbf{x}$ onto a one-dimensional latent space.

- The result yields a two-layer network that enjoys a universal approximation guarantee: any continuous $f$ can be approximated arbitrarily well by suitably chosen $\phi_{q,p}$ and $\Phi_q$.

- This decomposition attains universal approximation: given arbitrary $\varepsilon > 0$, one can choose continuous $\phi_{q,p}, \Phi_q$ so that $\|f - f_{\mathrm{KA}}\|_\infty < \varepsilon$.

## A.3 KOLMOGOROV–ARNOLD NETWORKS (KAN).

KAN generalises this construction into deep architectures. At layer $l$, let the input width be $n_l$ and the output width be $n_{l+1}$. We parameterise a matrix of univariate basis functions:

$$\Phi^{(l)} = \left\{ \phi_{j,i}^{(l)} \right\}, i = 1, \ldots, n_l, \, j = 1, \ldots, n_{l+1}, \tag{27}$$

and define the layer transform:

$$x_j^{(l+1)} = \sum_{i=1}^{n_l} \phi_{j,i}^{(l)}\big(x_i^{(l)}\big), \quad j = 1, \ldots, n_{l+1}. \tag{28}$$

Each $\phi_{j,i}^{(l)}$ is implemented as a cubic B-spline with learnable knot positions $\mu_{j,i}^{(l)}$ and widths $\sigma_{j,i}^{(l)}$. Concretely, given a projection of $x_i$:

$$\xi_i^{(l)} = \mathbf{a}_{j,i}^\top x_i^{(l)}, \tag{29}$$

the spline response is:

$$\phi_{j,i}^{(l)}(x_i^{(l)}) = B\big(\xi_i^{(l)}; \mu_{j,i}^{(l)}, \sigma_{j,i}^{(l)}\big), \tag{30}$$

where $B(\cdot; \mu, \sigma)$ denotes the standard cubic B-spline kernel. Stacking $L$ such layers yields:

$$x^{(L)} = \underbrace{\Phi^{(L-1)} \circ \cdots \circ \Phi^{(0)}}_{L \text{ layers}}(x^{(0)}), \tag{31}$$

which retains universal approximation and supports gradient-based training.

## A.4 B-SPLINE BASIS: DEFINITION AND PROPERTIES.

A cubic B-spline $B(u; \mu, \sigma)$ is defined piecewise by:

$$B(u) = \begin{cases} \frac{2}{3} - |u|^2 + \frac{1}{2}|u|^3, & |u| \leq 1, \\ \frac{1}{6}(2 - |u|)^3, & 1 < |u| \leq 2, \\ 0, & |u| > 2, \end{cases} \tag{32}$$

with $u = (\xi - \mu)/\sigma$. Its compact support $[-2\sigma, 2\sigma]$ and $C^2$-continuity make it a flexible yet efficient choice for learnable nonlinear transformations.

## A.5 GRAPH-AWARE KAN FOR SOURCE DETECTION.

To adapt KAN to graph data, we interleave node-wise spline lifts with a community-aware message-passing scheme. Let $G = (V, E)$, $|V| = n$, with adjacency matrix $\mathbf{A}$. For each node $v_i$ with initial feature $\mathbf{x}_i^{(0)}$:

**1) Kernel-driven feature lift.** We begin by mapping each node's initial feature to a latent coordinate and evaluating a bank of cubic B-spline kernels as univariate activations.

$$\mathbf{z}_i = W_z \mathbf{x}_i^{(0)}, \quad \xi_i = \mathbf{a}^\top \mathbf{z}_i, \tag{33}$$

$$g_{i,r} = B_r(\xi_i; \mu_r, \sigma_r), \, r = 1, \ldots, k. \tag{34}$$

This produces a $k$-dimensional kernel response $\mathbf{g}_i$ capturing localized nonlinear cues.

**2) Community-guided sparsification.** To encode mesoscale structure and reduce computational load, we restrict message passing to salient intra-community links identified by a community detector. Using Louvain clustering, we derive a binary mask $\mathbf{P}_{\text{comm}}$ that retains only the top-$k_{\max}$ intra-community edges per node. The effective adjacency is $\mathbf{A}^{\text{comm}} = \mathbf{A} \odot \mathbf{P}_{\text{comm}}$.

**3) Kernel-adaptive attention.** On the sparsified graph, we compute kernel-augmented edge scores and their softmax normalisation independently for each attention channel.

$$e_{ij}^{(q)} = \mathbf{a}_q^\top [\mathbf{z}_i \| \mathbf{z}_j] + \sum_{r=1}^{k} \beta_{q,r} (g_{i,r} + g_{j,r}), \tag{35}$$

$$\alpha_{ij}^{(q)} = \frac{\exp(\text{LeakyReLU}(e_{ij}^{(q)}))}{\sum_{j' \in \mathcal{N}_i^{\text{comm}}} \exp(\text{LeakyReLU}(e_{ij'}^{(q)}))}. \tag{36}$$

Thus, attention weights reflect both feature similarity and kernel-based propagation signals.

**4) Message passing and update.** Using these attentions, we diffuse kernel responses from neighbours and aggregate them channel-wise to obtain node-level summaries.

$$\tilde{g}_{i,r}^{(q)} = \sum_{j \in \mathcal{N}_i^{\text{comm}}} \alpha_{ij}^{(q)} g_{j,r}, \quad s_i^{(q)} = \sum_{r=1}^{k} w_{q,r} \tilde{g}_{i,r}^{(q)}, \tag{37}$$

$$\mathbf{S}_i = [s_i^{(1)}, \ldots, s_i^{(m)}]^\top, \tag{38}$$

followed by a channel-wise nonlinearity or a second KAN layer to produce $\mathbf{x}_i^{(1)}$.

This integration exploits KAN's universal approximation at the node level, while community-guided sparsity and kernel-adaptive attention ensure scalable, topology-sensitive diffusion modeling.

**5) Complexity and Parameter Counts.** Analysing a single GraphKAN layer highlights its efficiency:

$$\text{Ops} = O(nk + |E| k + nm k),$$

covering spline evaluations, sparse edge traversals, and channel-aggregation. Each layer includes:

$$\underbrace{d_{\text{in}} d_h}_{W_z} + \underbrace{d_h}_{\mathbf{a}} + \underbrace{3k}_{\mu,\sigma \text{ splines}} + \underbrace{mk}_{\beta} + \underbrace{m \cdot 2d_h}_{\mathbf{a}_q} + \underbrace{mk}_{w_{q,r}}, \tag{39}$$

scaling linearly in the number of kernels $k$ and channels $m$. This contrasts favorably with standard GNNs, whose parameter counts grow superlinearly when dense attention or deep MLP backbones are used.

**6) Training and Optimization.** We train GraphKAN end-to-end with Adam and a weighted cross-entropy loss:

$$\mathcal{L} = -\frac{1}{n} \sum_{i=1}^{n} (\omega y_i \log \hat{p}_{i,1} + (1 - y_i) \log \hat{p}_{i,0}) + \lambda \|\Theta\|_2^2, \tag{40}$$

where $\omega = (n - |s|)/|s|$ balances source vs. non-source classes, and $\lambda$ is an $\ell_2$ regulariser. We apply a cosine learning-rate schedule and dropout in each GraphKAN layer to stabilise training. Models converge within 1000 epochs on a single NVIDIA RTX 3090 Ti.

## A.6 DATASETS.

We evaluate our method across 12 benchmark datasets—six static graphs and six cascade-based networks. Summary statistics are provided in Table 6.

## A.7 MORE DETAILS ABOUT COMPARISON WITH SOTA METHODS.

Table 7 (5% state-missing) reveals consistent trends across twelve datasets and three metrics. Source-centrality methods (LPSI, EPA) attain respectable ACC but low F1, indicative of class imbalance (e.g., Football: ACC 0.80–0.82, F1 0.30–0.31). GCN-based models (GCNSI, SIGN, GCSSI,

| Id | Networks | $|V|$ | $|E|$ | $\langle k \rangle$ | $CC$ | Id | Datasets | #Users | #Links | #Cascades | Avg. Length |
|----|----------|-------|-------|---------------------|------|----|----------|--------|--------|-----------|-------------|
| 1 | Football | 115 | 613 | 10.66 | 0.40 | 7 | Christianity | 2897 | 35624 | 589 | 22.90 |
| 2 | Jazz | 198 | 2742 | 27.70 | 0.62 | 8 | Memetracker | 4709 | 209194 | 12661 | 16.24 |
| 3 | Facebook | 4039 | 88234 | 43.69 | 0.61 | 9 | Android | 9958 | 48573 | 679 | 33.30 |
| 4 | LastFM | 7624 | 27806 | 7.29 | 0.22 | 10 | Twitter | 12627 | 309631 | 3442 | 32.60 |
| 5 | Enron | 36692 | 183831 | 10.02 | 0.50 | 11 | Douban | 23123 | 348280 | 10602 | 27.14 |
| 6 | Github | 37700 | 289003 | 15.33 | 0.17 | 12 | Weibo | 46684 | 502400 | 18954 | 38.76 |

Table 6: Statistics of selected datasets. Datasets 1–6 summarize static network properties, while 7–12 report user-level cascades.

| Id | Datasets | Metrics | Methods | | | | | | | | | |
|----|----------|---------|------|-----|-------|------|-------|--------|------|--------|--------|------|
| | | | LPSI | EPA | GCNSI | SIGN | GCSSI | ResGCN | IVGD | SL-VAE | GIN-SD | Ours |
| | Football | ACC | 0.78 | 0.80 | 0.79 | 0.80 | 0.75 | 0.79 | 0.81 | 0.82 | 0.85 | **0.94** |
| | | F1 | 0.30 | 0.31 | 0.24 | 0.41 | 0.39 | 0.43 | 0.65 | 0.62 | 0.66 | **0.77** |
| | | AUC | 0.81 | 0.80 | 0.81 | 0.81 | 0.78 | 0.82 | 0.84 | 0.81 | 0.82 | **0.93** |
| | Jazz | ACC | 0.79 | 0.77 | 0.78 | 0.80 | 0.77 | 0.79 | 0.79 | 0.80 | 0.81 | **0.92** |
| | | F1 | 0.28 | 0.30 | 0.22 | 0.38 | 0.35 | 0.37 | 0.58 | 0.59 | 0.64 | **0.71** |
| | | AUC | 0.81 | 0.82 | 0.80 | 0.77 | 0.78 | 0.78 | 0.83 | 0.79 | 0.82 | **0.91** |
| | Facebook | ACC | 0.81 | 0.78 | 0.74 | 0.79 | 0.79 | 0.80 | 0.81 | 0.80 | 0.83 | **0.92** |
| | | F1 | 0.23 | 0.23 | 0.10 | 0.42 | 0.39 | 0.41 | 0.63 | 0.62 | 0.66 | **0.77** |
| Networks 1-6 | | AUC | 0.78 | 0.78 | 0.72 | 0.77 | 0.81 | 0.83 | 0.81 | 0.80 | 0.84 | **0.92** |
| | LastFM | ACC | 0.84 | 0.78 | 0.75 | 0.80 | 0.79 | 0.79 | 0.80 | 0.78 | 0.85 | **0.89** |
| | | F1 | 0.21 | 0.19 | 0.09 | 0.39 | 0.37 | 0.38 | 0.59 | 0.58 | 0.65 | **0.70** |
| | | AUC | 0.78 | 0.77 | 0.72 | 0.83 | 0.75 | 0.79 | 0.83 | 0.81 | 0.87 | **0.90** |
| | Enron | ACC | 0.80 | 0.80 | 0.71 | 0.76 | 0.78 | 0.79 | 0.81 | 0.78 | 0.81 | **0.90** |
| | | F1 | 0.19 | 0.22 | 0.07 | 0.36 | 0.35 | 0.37 | 0.57 | 0.54 | 0.63 | **0.71** |
| | | AUC | 0.80 | 0.78 | 0.74 | 0.78 | 0.75 | 0.81 | 0.82 | 0.81 | 0.86 | **0.89** |
| | Github | ACC | 0.77 | 0.76 | 0.71 | 0.79 | 0.81 | 0.80 | 0.82 | 0.79 | 0.85 | **0.90** |
| | | F1 | 0.18 | 0.16 | 0.08 | 0.33 | 0.34 | 0.36 | 0.57 | 0.57 | 0.58 | **0.69** |
| | | AUC | 0.78 | 0.78 | 0.69 | 0.78 | 0.80 | 0.81 | 0.84 | 0.81 | 0.82 | **0.91** |
| | Christianity | ACC | 0.79 | 0.75 | 0.77 | 0.78 | 0.78 | 0.80 | 0.77 | 0.82 | 0.85 | **0.91** |
| | | F1 | 0.22 | 0.23 | 0.11 | 0.35 | 0.36 | 0.38 | 0.53 | 0.52 | 0.63 | **0.70** |
| | | AUC | 0.79 | 0.78 | 0.72 | 0.79 | 0.76 | 0.81 | 0.80 | 0.79 | 0.88 | **0.88** |
| | Memetracker | ACC | 0.83 | 0.80 | 0.74 | 0.76 | 0.77 | 0.78 | 0.81 | 0.80 | 0.85 | **0.89** |
| | | F1 | 0.21 | 0.23 | 0.12 | 0.33 | 0.33 | 0.36 | 0.51 | 0.51 | 0.61 | **0.66** |
| | | AUC | 0.85 | 0.80 | 0.73 | 0.78 | 0.78 | 0.81 | 0.79 | 0.80 | 0.85 | **0.88** |
| | Android | ACC | 0.83 | 0.78 | 0.71 | 0.77 | 0.80 | 0.79 | 0.81 | 0.81 | 0.85 | **0.92** |
| | | F1 | 0.20 | 0.19 | 0.10 | 0.36 | 0.34 | 0.36 | 0.50 | 0.52 | 0.70 | **0.76** |
| Datasets 7-12 | | AUC | 0.80 | 0.76 | 0.78 | 0.76 | 0.79 | 0.81 | 0.82 | 0.81 | 0.88 | **0.94** |
| | Twitter | ACC | 0.81 | 0.80 | 0.70 | 0.74 | 0.79 | 0.80 | 0.79 | 0.82 | 0.82 | **0.87** |
| | | F1 | 0.19 | 0.22 | 0.09 | 0.30 | 0.31 | 0.34 | 0.46 | 0.45 | 0.49 | **0.60** |
| | | AUC | 0.86 | 0.80 | 0.75 | 0.73 | 0.76 | 0.81 | 0.77 | 0.82 | 0.84 | **0.87** |
| | Douban | ACC | 0.82 | 0.78 | 0.75 | 0.78 | 0.80 | 0.82 | 0.79 | 0.78 | 0.84 | **0.85** |
| | | F1 | 0.17 | 0.20 | 0.14 | 0.31 | 0.29 | 0.33 | 0.46 | 0.47 | 0.52 | **0.61** |
| | | AUC | 0.80 | 0.76 | 0.78 | 0.75 | 0.79 | 0.80 | 0.81 | 0.79 | 0.82 | **0.86** |
| | Weibo | ACC | 0.80 | 0.79 | 0.78 | 0.75 | 0.76 | 0.78 | 0.79 | 0.78 | 0.83 | **0.85** |
| | | F1 | 0.16 | 0.18 | 0.13 | 0.29 | 0.31 | 0.34 | 0.47 | 0.50 | 0.50 | **0.64** |
| | | AUC | 0.81 | 0.79 | 0.76 | 0.78 | 0.74 | 0.82 | 0.80 | 0.81 | 0.85 | **0.88** |

Table 7: Performance comparison of all evaluated methods across the twelve datasets, with the best results highlighted. The proportion of state-missing nodes is set to 5%.

ResGCN) provide only modest gains in F1 and AUC and underuse temporal and positional cues (e.g., Facebook F1 0.39–0.42). Propagation-aware baselines (IVGD, SL-VAE, GIN-SD) are the strongest among baselines, yet gains diminish with fragmented communities or partial observability (e.g., Twitter best baseline F1 0.49; Weibo 0.50; Android AUC 0.88; Memetracker 0.85).

Our method yields a more balanced profile, combining high ACC with stronger F1 and consistently higher AUC on both static graphs and cascades. Representative improvements include Football (F1 0.66 to 0.77; AUC 0.84 to 0.93), Jazz (F1 0.64 to 0.71; AUC 0.83 to 0.91), Facebook (ACC 0.83 to 0.92; F1 0.66 to 0.77), Android (F1 0.70 to 0.76; AUC 0.88 to 0.94), and Douban (F1 0.52 to 0.61; AUC 0.82 to 0.86). These gains arise from three components: (i) learnable B-spline activations that adapt to local regimes and emphasize early-spread signals, (ii) community-guided sparsification that suppresses spurious cross-community paths, and (iii) kernel-adaptive attention that weights edges beyond feature similarity.

Metric behavior aligns with imbalance: ACC generally exceeds F1, but the gap narrows for our method. On Github, F1 increases from 0.58 (best baseline) to 0.69 and AUC increases from 0.84 to 0.91. Improvements are broad and persist under 5% missing-state information.

## A.8 VISUALIZATION.

To provide deeper insight into the detection of rumor sources and clarify the analytical findings, we present the outcomes of various approaches in Fig. 6 and Fig. 7. The plots show that GraphKAN concentrates on true sources, yielding clearer propagation fronts than competing baselines. This visual representation supports direct comparisons across multiple methods and enhances the interpretability of the intricate propagation patterns that emerge.

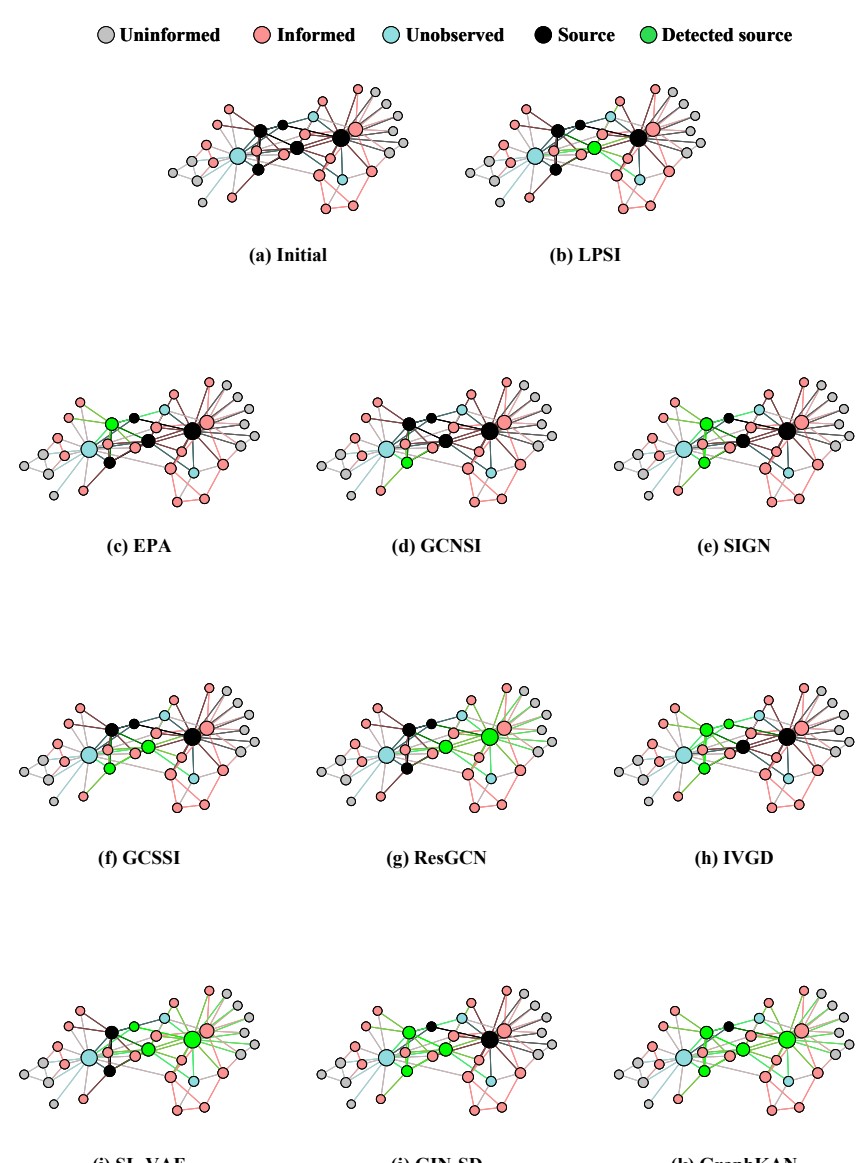

Figure 6: Visualization of source detection results on Karate network.

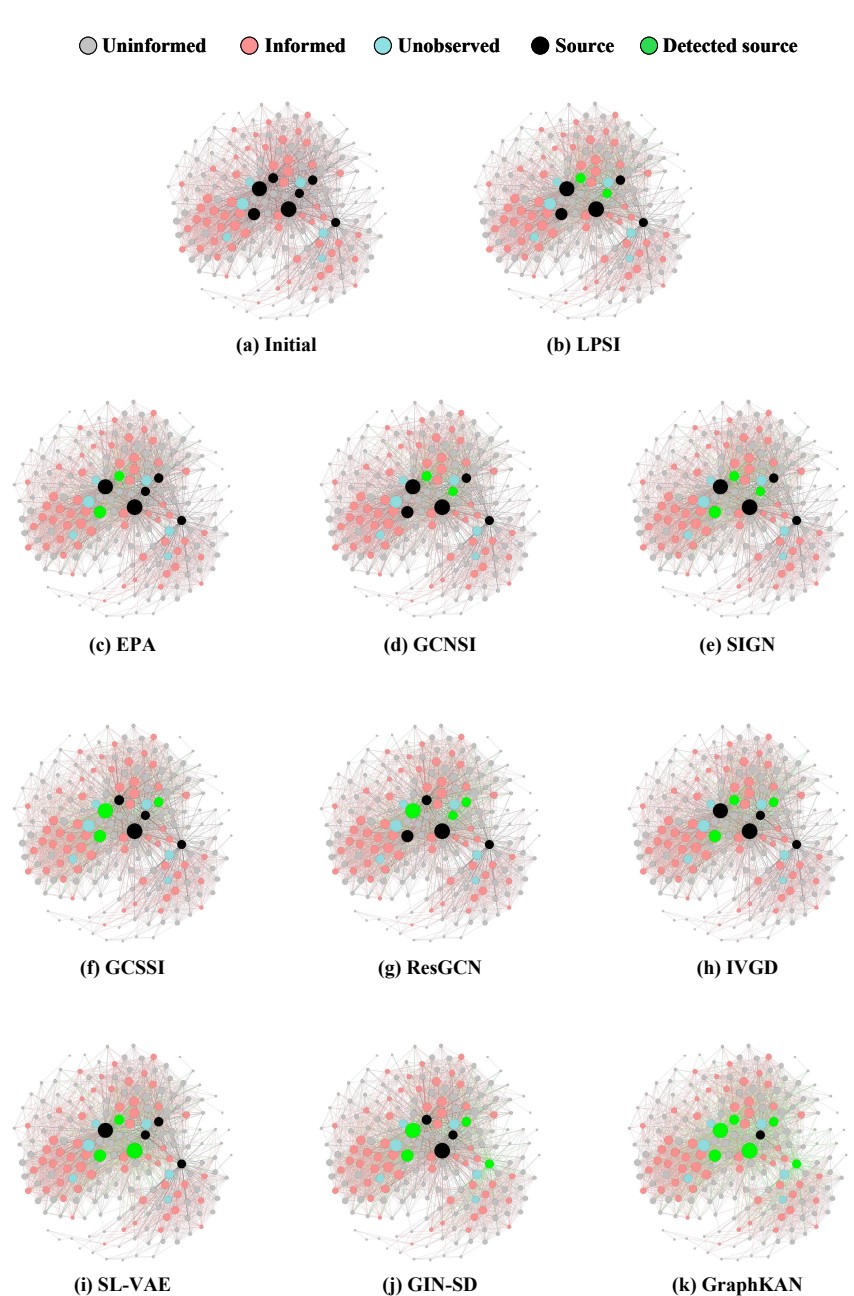

Figure 7: Visualization of source detection results on Jazz network.

