# OpenReview forum: "GraphKAN: An Efficient and Interpretable Kolmogorov-Arnold Graph Network for Source Detection"
_ICLR.cc/2026/Conference — ICLR 2026 Conference Withdrawn Submission_

### Official Review · Reviewer_QhaF · 2025-10-15

**Soundness:** 3
**Presentation:** 3
**Contribution:** 3
**Rating:** 4
**Confidence:** 4

**Summary:**

GraphKAN: An Efficient and Interpretable Kolmogorov–Arnold Graph Network for Source Detection, which capitalizes on Kolmogorov–Arnold Networks (KANs) by assigning learnable activation functions to edge weights.

**Strengths:**

1. motivation
- The design of community-guided sparsity and kernel-adaptive attention is well justified.
- a compact yet expressive message-passing mechanism.

2. Efficiency
- GraphKAN achieves lower memory footprint and faster inference while preserving high accuracy.

3. interpretability
- The visualized spline activations and node-importance scores provide interpretability evidence.

**Weaknesses:**

1. novelty
Several recent studies already combine KANs with GNNs [1,2,3]. This paper should more clearly articulate what is new: e.g., is the contribution primarily the community-guided sparsity, the kernel-adaptive attention, or a new theoretical formulation? A direct experimental comparison with prior Graph-KAN variants is missing.

2. evaluation
Many datasets are generated using the Independent Cascade (IC) model with only 2% missing states. While controllable, this setting is far cleaner than real social network data. The paper would benefit from testing robustness to higher noise or incomplete observations.

3. sensitivity
The effects of community granularity and sparsity parameters (e.g., top-k=5) are not systematically analyzed.

[1] Kiamari, M. et al. GKAN: Graph Kolmogorov-Arnold Networks. arXiv:2406.06470, 2024.
[2] Carlo Mastropietro et al. Kolmogorov-Arnold Graph Neural Networks. arXiv:2406.18354, 2024.
[3] Bresson, R. et al. KAGNNs: Kolmogorov-Arnold Networks meet Graph Learning. arXiv:2406.18380, 2024.

**Questions:**

1. Novelty
How exactly does GraphKAN differ from existing Graph-KAN or KAN-GNN architectures [1,2,3]? If the difference lies in the community-guided sparse aggregation, can you provide a direct comparison or ablation versus a baseline KAN-GNN without sparsity?

2. sensitivity.
How sensitive is performance to the choice of community size or top-k sparsity? Could you provide F1-vs-memory or top-k-vs-accuracy plots across datasets?

3. robustness
What happens when 20% or 50% of node timestamps are missing or randomly perturbed?

4. scalability
Could you report runtime trends on large graphs (e.g., >10^6 edges) and verify whether observed complexity matches the theoretical O(|E| log |V|) claim?

[1] Kiamari, M. et al. GKAN: Graph Kolmogorov-Arnold Networks. arXiv:2406.06470, 2024.
[2] Carlo Mastropietro et al. Kolmogorov-Arnold Graph Neural Networks. arXiv:2406.18354, 2024.
[3] Bresson, R. et al. KAGNNs: Kolmogorov-Arnold Networks meet Graph Learning. arXiv:2406.18380, 2024.

---

### Official Review · Reviewer_di5c · 2025-10-19

**Soundness:** 2
**Presentation:** 2
**Contribution:** 2
**Rating:** 2
**Confidence:** 4

**Summary:**

This paper introduces GraphKAN, an efficient and interpretable graph neural network for source detection. By integrating learnable B-spline activations from Kolmogorov–Arnold Networks, it enables flexible node feature transformations and adaptive edge attention. Experiments on twelve datasets show that GraphKAN surpasses existing methods in accuracy, efficiency, and interpretability.

**Strengths:**

1. The proposed GraphKAN is a new method for source detection that uses the strengths of Kolmogorov–Arnold Networks to overcome the limitations of fixed activations in traditional GNNs, achieving greater flexibility and expressiveness.
2. The effectiveness of GraphKAN is thoroughly evaluated on a wide range of real-world datasets, with additional interpretability analyses and ablation studies conducted to validate its design and demonstrate its promising performance.
3. The paper presents clear motivation and background for the problem it addresses, with a well-structured and logically organized presentation.

**Weaknesses:**

1. I think the novelty and contribution of this paper are insufficient for publication at ICLR. The key idea—extending KANs to graphs—is not new, as similar attempts have already appeared in many prior works. Moreover, the main contributions, proposed GraphKAN and its aggregation mechanism, are rather trivial. The paper simply combines several common components, such as state information, positional encodings, and channel-wise aggregation, without providing deeper theoretical analysis or intuition to justify these design choices. Overall, the model appears as a collection of loosely connected modules.
2. The experimental results in Table 1 show unusually large performance gains, raising concerns that the results may be too good to be true. In addition, the paper does not provide code for reproduction, nor does it report error statistics from multiple experimental runs.
3. The baselines used in the experiments are somewhat outdated, with only GIN-SD being a recent comparison. The paper should include more up-to-date baselines, such as [1] and [2].
4. The overall writing quality requires improvement. Please refer to the minor comments below.

[1] Cheng L, Zhu P, Gao C, et al. SDSI: Source Detection in Structurally Incomplete Social Networks[J]. IEEE Transactions on Network Science and Engineering, 2024.

[2] Ali S S, Rastogi A, Anwar T, et al. Generalized Local Prominence for Source Detection in Real-World Rumor Networks[J]. IEEE Transactions on Knowledge and Data Engineering, 2025.

**Minor comments:**
(1) Figure 1 lacks sufficient description and explanation in the main paper, making it difficult to understand its intended message.
(2) The mathematical notation is inconsistent. For example, Equation (1) uses $T, U$, and $P$ to denote different types of information, while later sections switch to $X^1, X^2, X^3$. Unify these symbols and follow the ICLR template conventions for representing vectors and matrices.
(3) The layout on pages 8 and 9 is poorly formatted, with figures and tables clustered together, making it hard to read.

**Questions:**

1. Please begin by responding to the Weaknesses part.
2. Could you elaborate on the theoretical advantages or underlying intuition of GraphKAN in terms of its expressive power or source detection capability compared with previous methods?
3. Please explain why the results in Table 1 show such a significant improvement, and also report the standard deviations across multiple runs, as mentioned in the weaknesses section.

---

### Official Review · Reviewer_KtLa · 2025-10-26

**Soundness:** 3
**Presentation:** 3
**Contribution:** 2
**Rating:** 4
**Confidence:** 4

**Summary:**

GraphKAN is an efficient and interpretable Kolmogorov-Arnold graph network for source detection (e.g., rumor tracing). It aims to address issues existing in current GCN-based source detection methods, such as the waste of non-embedding parameters, reliance on fixed activation functions, and poor interpretability.

**Strengths:**

1.Across twelve real-world datasets, GraphKAN consistently outperforms state-of-the-art baseline methods in metrics including Accuracy (ACC), F1-score, and Area Under the ROC Curve (AUC). It can detect source nodes in graphs more accurately, establishing a new performance benchmark for source detection tasks. Particularly when dealing with complex network structures and different types of datasets (static networks and cascade datasets), it demonstrates stable and excellent detection capabilities.

2.By exposing interpretable intermediate representations through learnable basis functions (univariate B-spline functions), GraphKAN enables the analysis of learned activation patterns and attention weights, clarifying the relative importance of each node in the source detection process. This breaks the black-box nature of traditional GCN methods, allowing researchers and users to clearly understand the model's decision-making process, enhancing trust in the model's results. It holds significant importance in scenarios requiring traceability and verification of detection results (e.g., responsibility determination for rumor source tracing).

3.The adoption of a sparsity-aware neighborhood aggregation strategy based on community clustering reduces unnecessary message passing and lowers computational complexity.

**Weaknesses:**

1.Over the past year, there have been numerous studies on the integration of graphs and KAN networks. Among them, peer-reviewed works include [1], [2], [3], etc., and non-peer-reviewed ones include [4], [5]. It is noticeable that there are multiple works named "GraphKAN". Personally, the authors should consider using a different name for their work.

2.The RELATED WORK section of the manuscript lacks a summary of studies combining graphs and KAN networks. It fails to explain why previous works on the integration of graphs and KAN networks cannot be directly applied to source detection tasks. Additionally, it does not highlight the differences between the authors' work and previous studies.

3.Although hyperparameters are provided in the Experimental Settings section, there is a lack of hyperparameter analysis. For instance, k-max in Equation 14 should be subjected to corresponding hyperparameter analysis. Moreover, the ablation study is also deficient in variants of community detection algorithms.

[1] GraphKAN: An Efficient Graph Kolmogorov Arnold Networks for Traffic Forecasting

[2] G-KAN: Graph Kolmogorov-Arnold Network for Node Classification Using Contrastive Learning

[3] KAGAT: Kolmogorov-Arnold Graph Attention Network

[4] GraphKAN: Graph Kolmogorov Arnold Network for Small Molecule-Protein Interaction Predictions

[5] GraphKAN: Enhancing Feature Extraction with Graph Kolmogorov Arnold Networks

**Questions:**

1.It is necessary to summarize previous works that combine graphs with KAN, and clarify the advantages of the proposed work compared to these existing studies.

2.In the experimental section, it is advisable to supplement comparisons with one or two works that integrate graphs and KAN.

3.It is expected to add hyperparameter analysis and ablation experiments related to community detection algorithms.

---

### Official Review · Reviewer_G44r · 2025-10-27

**Soundness:** 2
**Presentation:** 3
**Contribution:** 3
**Rating:** 6
**Confidence:** 3

**Summary:**

Thank you for the opportunity to review this paper. The paper presents GraphKAN, a graph neural network based on Kolmogorov–Arnold Networks for efficient and interpretable source detection such as rumor tracing. It replaces fixed activation functions in conventional GCNs with learnable B-spline functions to capture flexible, localized nonlinear patterns, and introduces a community-guided sparse aggregation mechanism that focuses message passing within key regions of the graph. Experiments on twelve real-world datasets show that GraphKAN achieves higher accuracy, lower computational cost, and better interpretability than existing approaches.

**Strengths:**

1. The presentation and organization quality overall are good and clear.
2. The proposed framework seems reasonable.
3. The experiments are comprehensive and complete.
4. The literatures are comprehensive and complete.

**Weaknesses:**

1. Some parts of presentation can be improved better.
2. Experimental results may not be convincing.
3. The code does not release. It may be difficult to reproduce results.

**Questions:**

It is unclear what the key challenge is in applying the KAN model to the graph source detection problem. Specifically, why is incorporating graph topology into KAN difficult, and why can’t conventional GNN frameworks be directly adapted for this purpose?

The novelty of this paper is somewhat unclear. The authors may consider emphasizing the unique design components or mechanisms introduced to address the aforementioned challenges.

The experimental results lack information on standard deviations. Were the experiments repeated multiple times? Without this, the reported improvements may not be statistically significant.

The paper should provide descriptive statistics of the datasets used, such as the number of nodes, edges, and cascades, to help readers better understand the experimental setup. Are these graph large enough to test scalability?

---

### Note · Authors · 2025-11-25

I have read and agree with the venue's withdrawal policy on behalf of myself and my co-authors.